# Trajectories of CD4^+^/CD8^+^ T-Cells Ratio 96 Weeks after Switching to Dolutegravir-Based Two-Drug Regimens: Results from a Multicenter Prospective Cohort Study

**DOI:** 10.3390/v14112315

**Published:** 2022-10-22

**Authors:** Lucia Taramasso, Antonio Falletta, Elena Ricci, Giancarlo Orofino, Nicola Squillace, Barbara Menzaghi, Giuseppe Vittorio De Socio, Chiara Molteni, Giovanni Francesco Pellicanò, Roberto Gulminetti, Giordano Madeddu, Eleonora Sarchi, Francesca Vichi, Benedetto Maurizio Celesia, Paolo Bonfanti, Antonio Di Biagio

**Affiliations:** 1Infectious Diseases Unit, San Martino Policlinico Hospital, IRCCS for Oncology and Neurosciences, 16132 Genoa, Italy; 2Department of Health Sciences (DiSSal), University of Genova, 16132 Genova, Italy; 3Fondazione A.S.I.A. Onlus, 20100 Milan, Italy; 4Division I of Infectious and Tropical Diseases, ASL Città di Torino, 10149 Turin, Italy; 5Infectious Diseases Unit ASST-MONZA, San Gerardo Hospital, University of Milano-Bicocca, 20900 Monza, Italy; 6Unit of Infectious Diseases, ASST della Valle Olona, 21052 Busto Arsizio, Italy; 7Infectious Diseases Unit, Department of Internal Medicine 2, “Santa Maria della Misericordia” General Hospital, 06129 Perugia, Italy; 8Infectious Diseases Unit, Ospedale A. Manzoni, 23900 Lecco, Italy; 9Unit of Infectious Diseases, Department of Human Pathology of the Adult and the Developmental Age “G. Barresi”, University of Messina, 98124 Messina, Italy; 10Department of Medical Sciences and Infectious Diseases, Fondazione IRCCS Policlinico San Matteo, 27100 Pavia, Italy; 11Unit of Infectious Diseases, Department of Medical, Surgical and Experimental Sciences, University of Sassari, 07100 Sassari, Italy; 12Infectious Diseases Unit, S. Antonio e Biagio e Cesare Arrigo Hospital, 15121 Alessandria, Italy; 13Infectious Diseases Unit, Santa Maria Annunziata Hospital, 50012 Bagno a Ripoli, Italy; 14Unit of Infectious Diseases, University of Catania, ARNAS Garibaldi Catania, 95123 Catania, Italy

**Keywords:** dolutegravir, two-drug regimen, three-drug regimen, ART, immune recovery, CD4/CD8 ratio

## Abstract

The aim of the present study was to evaluate CD4/CD8 dynamics in patients on dolutegravir (DTG)-based two-drug regimens (2DRs) and compare them with DTG-containing triple-drug regimens (3DRs). A prospective observational study was performed in the context of the SCOLTA cohort. Experienced PWH with HIV-RNA < 50 copies/mL were included if they were on the DTG-2DR, the DTG + tenofovir/emtricitabine (TDF/FTC) regimen, the DTG + tenofovir alafenamide (TAF)/FTC regimen, or the DTG + abacavir/lamivudine (ABC/3TC) regimen; they were followed-up for at least one year. A total of 533 PWH were enrolled, 120 in the DTG + 3TC group, 38 in the DTG + protease inhibitors (PI) group, 67 in the DTG + rilpivirine (RPV) group, 49 in the DTG + TDF/FTC group, 27 in the DTG + TAF/FTC group, and 232 in the DTG + ABC/3TC group. After one year, the CD4/CD8 ratio significantly increased in the PWH treated with DTG + 3TC (+0.08 ± 0.26), DTG + TDF/FTC (+0.1 ± 0.19), and DTG + ABC/3TC (+0.08 ± 0.25). At two years, the CD4/CD8 increase was confirmed for PWH on DTG + TDF/FTC (+0.16 ± 0.28) and DTG + ABC/3TC (+0.1 ± 0.3). In the SCOLTA cohort, PWH on 2DRs experienced a CD4/CD8 increase only in the DTG + 3TC group. Controlled studies with longer follow-up will clarify the long-term immunological and clinical impacts of DTG-2DR.

## 1. Introduction

The introduction of antiretroviral therapy (ART) has improved health of people living with HIV (PWH), prolonged their lives, and substantially reduced the risk of HIV-1 transmission [1]. Although ART has changed the clinical history of HIV/AIDS, the requirement for ongoing treatment poses multiple challenges for PWH [2]. In this perspective, new simplified therapeutic options with fewer drug regimens have been explored in optimization strategies with the aim of minimizing unnecessary drug exposure and side effects [3]. Different Dolutegravir (DTG)-based two-drug regimens (2DRs) have been studied so far [3,4]. However, persistent inflammation and immune activation has been described among PWH [5] and the possible implications of 2DRs on long-term residual inflammation are still under study. A low CD4/CD8 ratio could be interpreted as the epiphenomenon of a dysregulation and immunological activation of patient’s immune system with hyperinflammation and immunosenescence [5], and has been associated with non-AIDS-defining events, mortality, and premature aging. This has been shown to also be the case among a subset of PWH with virological control and CD4 reconstitution; despite recent data covering very large cohorts, researchers have failed to find a significant independent association between CD4/CD8 ratio and all-cause mortality [6,7]. In the past, the use of protease inhibitor (PI)-based dual therapies had been reported as a risk factor for lack of CD4/CD8 restoration [8]. However, little is known on the CD4/CD8 trajectories in course of the modern DTG-based 2DR.

DTG/lamivudine (3TC) 2DR has been studied in ART-naïve and ART-experienced patients in the two large, randomized GEMINI and TANGO trials [9,10]. At 96 weeks, both demonstrated non-inferiority of DTG-2DR compared with triple-drug regimens, and CD4/CD8 ratio and other inflammatory markers (blood D-dimer, serum hs-C reactive protein, serum InterLeukin-6, sCD14, sCD163) had similar trends following 2DR and 3DR. Moreover, SWORD-1 and SWORD-2 trials showed no changes in inflammatory markers at 48 weeks after switching from a suppressive ART regimen to DTG/rilpivirine (RPV) 2DR [4]. Data on efficacy and safety of DTG-based 2DR have also been confirmed in observational studies [11,12] and significant increases in the absolute CD4^+^ T-cell count and CD4/CD8 ratio have been reported at both 96 and 144 weeks in virologically suppressed PWH switching to DTG/3TC [13]. The aim of the present study is to evaluate CD4/CD8 dynamics among PWH on DTG-based 2DR in a large multicenter prospective cohort study and compare them with DTG-based triple-drug therapies (3DR).

## 2. Materials and Methods

We analyzed data from the SCOLTA (Surveillance Cohort Long-Term Toxicity Antiretrovirals) prospective database. The SCOLTA project is a multicenter observational study which started in 2002 and prospectively follows PWH who initiate new antiretroviral drugs in real-life settings [14,15,16]. The SCOLTA project uses an online pharmacovigilance program and involves 25 Italian Infectious Disease Centers. Both ART-naïve and ART-experienced patients can be included in SCOLTA if they are >18 years and agree to the study terms on entry. Clinical data collected include sex, age, ethnicity, weight, height, Center for Disease Control and Prevention (CDC) stage, and previous ART history. Laboratory data include HIV-RNA, CD4^+^ T-cell count, CD4/CD8 ratio, and biochemical data, and are prospectively collected in anonymous form in a central database every six months. For the present study, we included ART-experienced PWH with HIV-RNA < 50 copies/mL who switched to DTG enrolled from July 2014 to June 2021 (date of last extraction) and had at least one-year follow-up after enrolment in SCOLTA. PWH were included if they were on a DTG-based 2DR with 3TC, PI, or RPV, or were on a DTG-based 3DR with tenofovir/emtricitabine (TDF/FTC), tenofovir alafenamide (TAF)/FTC, or abacavir/lamivudine (ABC/3TC).

The original study protocol was approved on 18 September 2002, and a new protocol amendment was approved on 13 June 2013 by the coordinating center at Hospital “L. Sacco”, University of Milan, and thereafter by all participating centers. Written consent for study participation was obtained from all participants, and the study was conducted in accordance with the ethical standards laid down in the 1964 Declaration of Helsinki and its later amendments and by Italian national laws.

The primary objective was to evaluate CD4/CD8 trajectories in different DTG-based 2DRs. The secondary objectives were to assess the difference between CD4/CD8 ratio under 2DRs and 3DRs and investigate the risk factors for failing to achieve CD4/CD8 ≥ 1.0 [17,18].

### Statistical Analysis

Data were described using mean and standard deviation (SD) for normally distributed continuous variables; median and interquartile range (IQR) for non-normally distributed continuous variables; frequency and percentage (%) for categorical and ordinal variables. CD4/CD8 ratio change from baseline was investigated through the paired t-test within DTG-2DRs and 3DRs, while comparison of CD4/CD8 trend among different regimens was investigated using a mixed linear model including terms for interaction between regimen and follow-up visit. The Cox proportional hazards model was used for estimating the hazard ratio (HR) and 95% confidence interval (95% CI) of failure to achieve CD4/CD8 ratio ≥ 1.0. The multivariable model included variables that were statistically significant at univariate analysis. The significance level was defined at *p* < 0.05.

## 3. Results

A total of 533 ART-experienced PWH were included in the analysis and followed for a median period of 24 months (IQR 13–49). At baseline, the current ART regimen was a DTG-2DR in 225 (42.2%) PWH: 120 (22.5%) were on DTG + 3TC, 38 (7.1%) were on DTG + PI, and 67 (12.6%) were on DTG + RPV. A total of 308 (57.8%) PWH were on 3DR: 49 (9.2%) were on DTG + TDF/FTC, 27 (5.1%) were on DTG + TAF/FTC, and 232 (43.5%) were on DTG + ABC/3TC. Median time on current regimen was 24 months (IQR 13–49).

The main characteristics of the study population at baseline are shown in Table 1 and Appendix A.

Briefly, 27.2% were female, 92.3% of Caucasian origin, with mean age of 51.0 (±SD 10.9) years; 70.4% acquired HIV through unprotected sex (men who have sex with men (31.0%) and heterosexual (39.4%)); 17.6% acquired HIV through injection drug use (IDU). HBsAg was positive in 2.8% of study participants, and 24.5% had positive HCV serology. A total of 44 PWH (8.3%) had impaired renal function (eGFR < 60 mL/min/1.73 m^2^), 29 (5.4%) had diabetes, and 126 (23.6%) had hypertension.

As for immunological parameters, median CD4^+^ T-cell count was 688 (IQR 466–908) cells/mm^3^, and CD8^+^ T-cell count was 816 (IQR 620–1120) cells/mm^3^ at study entry, with mean CD4/CD8 ratio 1.04 (±0.64) in DTG-2DR, 0.74 (±0.47) in DTG + TDF/FTC, 0.88 (±0.64) in DTG + TAF/FTC, and 0.86 (±0.50) in DTG + ABC/3TC (*p* = 0.0001).

At one-year follow up, CD4/CD8 ratio changed in PWH treated with DTG + 3TC (+0.08 ± 0.26, *p* = 0.004), DTG + TDF/FTC (+0.10 ± 0.19, *p* = 0.002), and DTG + ABC/3TC (+0.08 ± 0.25, *p* < 0.0001) (Table 2). At two-year follow up, CD4/CD8 increase was confirmed to be significant only among those taking 3DR with DTG + TDF/FTC (+0.16 ± 0.28, *p* = 0.003) and DTG + ABC/3TC (+0.10 ± 0.30, *p* < 0.0001), while no significant change was observed among any of the DTG-2DR-prescribed PWH (Table 2, Figure 1). The CD4/CD8 increase was driven by a significant increase in CD4^+^ T cells, while CD8^+^ T cells remained stable or slightly increased during the follow-up but did not achieve significant reductions in any treatment groups.

After the exclusion of people with comorbidities or conditions potentially affecting the immune recovery (i.e., cirrhosis, autoimmune diseases and/or malignancies, previous CVD events, *N* = 59), we found that the CD4/CD8 ratios showed similar increases to those of the whole study population.

### Predictors of CD4/CD8 Restoration to ≥1.0

Among 533 study participants, 342 (64.2%) had a CD4/CD8 ratio lower than 1.0 at baseline. This proportion was significantly higher among PWH on 2DR (43.6%) than among those on DTG + TDF/FTC (26.5%), DTG + TAF/FTC (25.9%), or DTG + ABC/3TC (31.5%) (*p* = 0.01).

Among them, 101/342 (29.5%) achieved a CD4/CD8 ratio ≥ 1.0 over a median time of 412 days (IQR 194–713), while the overall observation time was 574 days (IQR 305–1224).

Table 3 summarizes the characteristics of PWH according to achievement of CD4/CD8 ratio ≥ 1.0 during the study period. PWH who did not reach a CD4/CD8 ratio ≥ 1.0 were older, more frequently male, with HCV co-infection, CDC stage C, renal function impairment (eGFR ≤ 60), or lower CD4/CD8 ratio at baseline.

The variables associated with CD4/CD8 ratio normalization in the univariate model are shown in Table 4. CDC stage C, integrase inhibitors (INSTI) use in previous regimen, and lower baseline CD4/CD8 ratio were confirmed as determinants associated with an increased risk of failure in CD4/CD8 ratio recovery.

The determinants of CD4/CD8 ratio recovery in patients on 2DRs are shown in Table 5. In this group, CD4/CD8 ratio failed to normalize among PWH with lower baseline CD4/CD8, while DTG + 3TC was associated with higher probability of DC4/CD8 ratio recovery compared with DTG + RPV.

## 4. Discussion

In recent years, triple-drug antiretroviral regimens have represented the standard of care for controlling and managing HIV viral replication.

Nowadays, the increasing prevalence of aging-related comorbidities requires simplified therapeutic options with fewer drug regimes in order to minimizing drug exposure and side effects. For this purpose, new ART strategies such as 2DR have been explored, showing tolerability, fewer adverse events, and non-inferior efficacy compared with 3DR [15,19].

However, data about the impact of 2DR on immune activation and systemic inflammation are still scant. During treated HIV infection, immune activation and biomarkers of inflammation improve within the first years, but residual immune activation may persist [7,20], with high CD8^+^ T-cell count and low CD4/CD8 ratio [21,22]. It is notable that low CD4/CD8 ratios have been associated with increased risk of non-AIDS-related events and death [23].

In the present study, we observed that PWH on DTG-2DR experienced CD4/CD8 increase at one year follow-up only in the DTG + 3TC group, while no significant change was observed in DTG + RPV and DTG + PI. Data from other published real-life cohort studies confirm our results. Monsalvo et al. reported an increase in CD4/CD8 ratio by 3% (+0.02, IQR −0.07, +0.09, *p* = 0.07) at 48 weeks in 245 PWH and HIV-RNA < 50 copies/ml who had switched to 2DRs [24]. The slope of rise was related to baseline values and previous HIV-related factors, but unlike our results, there were no differences in the CD4/CD8 ratios among the type of dual regimen. In more detail, in our cohort, CD4/CD8 increase was driven by a significant increase in CD4^+^ T cells, while CD8^+^ T cells remained stable or slightly increased during the follow-up but did not achieve any significant modifications in any treatment groups.

These data contrast with other published studies, which reported a slight increase in CD8^+^ T lymphocytes count after switching to dual therapy or monotherapy. Mussini et al. ran a study comparing the dynamics of CD4/CD8 ratio in people with undetectable HIV-RNA who underwent a switch to 2DRs to a control group remaining on 3DRs; they observed a stabilization of the CD4/CD8 ratio related to no significant difference in the CD4^+^ T-cell count trajectory but to a specific increase in CD8^+^ T lymphocyte count. This finding was possibly linked to residual viremia triggering CD8^+^ T-cell activation and proliferation [8]. However, it is necessary to clarify that 2DRs in that study were different from ours, not consisting of DTG-based 2DRs.

Many studies in the literature have explored the trajectories of other immune-activation markers after simplification treatment from 3DRs to 2DRs, with conflicting opinions.

Vassallo et al. showed how ART simplification to 2DR was associated with macrophage activation despite sustained virological control, with a time of onset during the first year following the switch [25]. The explanation to this immune activation is still unclear but some authors hypothesize a diminished drug pressure on tissue and cellular reservoirs [26].

Regarding predictors of CD4/CD8 ratio restoration, data for 2DR remain scant.

In our work, CDC stage C, INSTI in the previous regimen, and lower baseline CD4/CD8 ratio were associated with lack of CD4/CD8 ratio restoration, while in 2DR the only predictive factor for CD4/8 ratio normalization seemed the baseline of this ratio, although the low number of patients on these regimens may have prevented us from finding other statistically significant determinants. Additionally, PWH in DTG + RPV seemed to achieve a CD4/CD8 ratio ≥ 1 less frequently, possibly due to comorbidities and a more complex clinical picture at baseline from patients in this regimen.

Our study is limited by its observational model and the absence of randomized design in addition to low power due to sample size and two-year follow-ups. Owing to the observational nature of the study, PWH were not randomly assigned to drug therapies, and confounding bias could have occurred, based on clinicians’ decisions to prescribe different drug associations in patients with different baseline characteristics.

Another limitation in our study resides in the lack of data about the co-existence and the effect of other life-long viral coinfections, such as cytomegalovirus and Epstein–Barr on CD4^+^ and CD8^+^ T-cell count dynamics in our cohort of individuals during ART [27,28,29].

In conclusion, PWH on DTG + 3TC experienced a significant improvement of CD4/CD8 ratio even after a long history of ART experience. Controlled studies with longer follow-up will clarify the clinical impact of DTG-2DR.

## Figures and Tables

**Figure 1 viruses-14-02315-f001:**
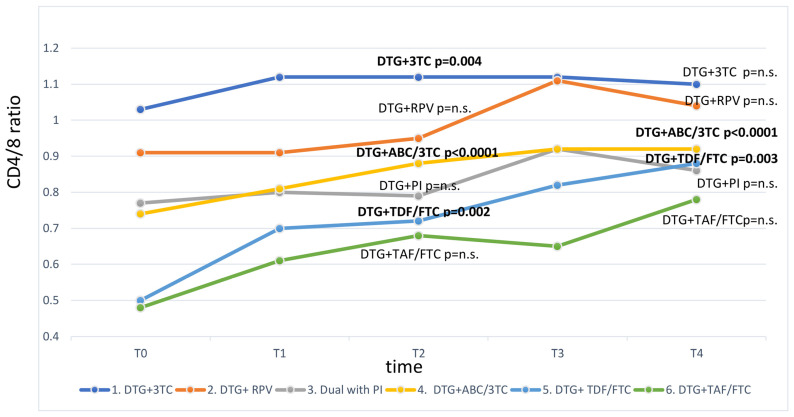
CD4 and CD8 T cells and CD4/CD8 ratio dynamics in patients in treatment with Dolutegravir. Values that significantly changed compared to baseline (*p* < 0.05) are indicated in bold; *p* = n.s.—p value not significant. 3TC—lamivudine; ABC—abacavir; DTG—Dolutegravir; FTC—emtricitabine; PI—protease inhibitors; RPV—rilpivirine; TAF—tenofovir alafenamide; TDF—tenofovir disoproxil fumarate; T0—baseline; T1—six-month follow-up; T2—one-year follow-up; T3—eighteen-month follow-up; T4—two-year follow-up.

**Table 1 viruses-14-02315-t001:** Baseline characteristics of 533 people living with HIV (PWH) enrolled in Dolutegravir (DTG) cohort.

	Total	DTG + 3TC	DTG + PI	DTG + RPV	DTG + TDF/FTC	DTG + TAF/FTC	DTG + ABC/3TC	*p*
	N = 533	N = 120 (22.5%)	N = 38 (7.1%)	N = 67 (12.6%)	N = 49 (9.2%)	N = 27 (5.1%)	N = 232 (43.5%)	
**Age (years), mean ± SD**	51.0 ± 10.9	50.7 ± 11.1	53.4 ± 8.6	53.4 ± 11.1	48.9 ± 10.4	50.6 ± 12.3	50.4 ± 10.8	0.13
**Sex F, n (%)**	145 (27.2)	28 (23.3)	6 (15.8)	27 (40.3)	14 (28.5)	7 (25.9)	63 (27.1)	0.10
**Caucasian ethnicity, n (%)**	492 (92.3)	111 (92.5)	34 (89.5)	63 (94.0)	48 (97.9)	23 (85.1)	213 (91.8)	0.42
**Risk Factor for HIV acquisition, n (%)**								
Heterosexual	210 (39.4)	49 (40.8)	13 (34.2)	29 (43.2)	16 (32.6)	9 (33.3)	94 (40.5)	<0.0001
MSM	165 (31.0)	49 (40.8)	11 (29.0)	16 (23.8)	9 (18.3)	12 (44.4)	68 (29.3)	
IDU	94 (17.6)	14 (11.6)	6 (15.8)	20 (29.8)	10 (20.4)	6 (22.2)	38 (16.3)	
Other	64 (12.0)	8 (6.6)	8 (21.0)	2 (2.9)	14 (28.5)	0 (0.0)	32 (13.7)	
**BMI (Kg/m^2^), n (%)**								
≤25.0	269 (56.3)	52 (51.5)	20 (58.8)	34 (56.7)	26 (57.8)	9 (37.5)	128 (59.8)	0.31
>25.0–30.0	168 (35.2)	39 (38.6)	13 (38.2)	21 (35.0)	12 (26.7)	11 (45.8)	72 (33.6)	
>30.0	41 (8.6)	10 (9.9)	1 (2.9)	5 (8.3)	7 (15.6)	4 (16.7)	14 (6.5)	
**HBsAg positive, n (%)**	14 (2.8)	3 (2.5)	0 (0.0)	0 (0.0)	3 (6.8)	4 (15.3)	4 (1.8)	0.0007
**HCV coinfection, n (%)**	127 (24.5)	22 (18.8)	8 (21.1)	22 (33.8)	14 (30.4)	6 (23.0)	55 (24.1)	0.29
**Cirrhosis**	6 (1.1)	0	0	1 (1.5)	3 (1.3)	0	2 (7.4)	0.03
**CDC stage, n (%)**								
A	262 (49.2)	71 (59.2)	20 (52.6)	28 (41.8)	24 (49.0)	13 (48.2)	106 (45.7)	0.004
B	166 (31.1)	37 (30.8)	8 (21.0)	29 (43.3)	13 (26.5)	9 (33.3)	70 (30.2)	
C	105 (19.7)	12 (10.0)	10 (26.3)	10 (14.9)	12 (24.5)	5 (18.5)	56 (24.1)	
**CD4 (cells/mL), median (IQR)**	688 (466–908)	735 (594–1001)	644 (407–904)	684 (490–841)	555 (300–867)	652 (429–869)	656 (455–900)	0.01
**CD8 (cells/mL), median (IQR)**	816 (620–1120)	786 (614–1018)	778 (618–1193)	816 (614–1120)	922 (615–1177)	838 (744–1033)	834 (620–1175)	0.40
**CD4/CD8, mean ± SD**	0.92 ± 0.58	1.13 ± 0.63	0.90 ± 0.67	0.95 ± 0.62	0.74 ± 0.47	0.88 ± 0.64	0.86 ± 0.50	0.0003
**Nadir CD4 (cells/mL), median (IQR) (N = 86)**	263 (171–368)	328 (211–447)	-	201 (58–263)	-	250 (99–294)	214 (161–311)	0.06
**Diabetes, n (%)**	29 (5.4)	5 (4.17)	1 (2.6)	2 (2.9)	5 (10.2)	3 (11.1)	13 (5.6)	0.35
**Hypertension, n (%)**	126 (23.6)	31 (25.8)	8 (21.0)	13 (19.4)	7 (14.2)	5 (18.5)	62 (26.7)	0.40
**Other cardiovascular diseases, n (%)**	30 (5.6)	9 (7.5)	6 (15.8)	4 (6.0)	8 (3.4)	2 (4.1)	1 (3.7)	0.06
**Malignancies, n (%)**	15 (2.8)	1 (0.8)	1 (2.6)	5 (7.5)	6 (2.6)	2 (4.1)	0	0.15
**Auto-immune diseases, n (%)**	9 (1.7)	1 (0.8)	2 (5.3)	0	5 (2.2)	1 (2.0)	0	0.37
**eGFR < 60 (mL/min), n (%)**	44 (8.2)	9 (7.5)	3 (8.1)	8 (11.9)	3 (6.1)	3 (11.1)	18 (7.7)	0.86

3TC—lamivudine; ABC—abacavir; BMI—body mass index; ATV—atazanavir; CD4—CD4^+^ T cells; cART—combined antiretroviral therapy; CDC—Center for Disease Control and Prevention; DRV; darunavir; DTG—Dolutegravir; eGFR—estimated glomerular filtration rate; EFV—efavirenz; EVG—elvitegravir; FTC—emtricitabine; HBsAg—hepatitis B surface antigen; HCV—hepatitis C virus; IDU—injecting drug use; INSTI—integrase strand transfer inhibitors; IQR—interquartile range; MSM—men who have sex with men; NNRTI—non-nucleoside reverse transcriptase inhibitors; NRTI—nucleoside reverse transcriptase inhibitors; NVP—nevirapine; PI—protease inhibitors; RAL—raltegravir; RPV—rilpivirine; SD—standard deviation; TAF—tenofovir alafenamide; TDF—tenofovir disoproxil fumarate.

**Table 2 viruses-14-02315-t002:** CD4^+^ and CD8^+^ T cells and CD4/CD8 ratio dynamics in patients in treatment with Dolutegravir (DTG).

	DTG + 3TC			DTG + PI			DTG + RPV			DTG + TDF/FTC			DTG + TAF/FTC			DTG + ABC/3TC		
	N	Mean	SD	N	Mean	SD	N	Mean	SD	N	Mean	SD	N	Mean	SD	N	Mean	SD
**CD4**																		
T1-T0	*120*	**45**	**183**	*38*	−1	156	*67*	−1	192	*49*	82	313	*27*	12	170	*232*	**38**	**198**
T2-T0	*93*	32	237	*36*	14	179	*51*	19	174	*43*	**89**	**202**	*16*	−39	174	*199*	**76**	**218**
T3-T0	*68*	55	231	*32*	69	215	*35*	25	224	*44*	**146**	**333**	*7*	54	314	*160*	**94**	**259**
T4-T0	*40*	**127**	**268**	*30*	39	179	*32*	8	219	*36*	**98**	**261**	*5*	44	330	*142*	**145**	**273**
**CD8**																		
T1-T0	*120*	0	200	*38*	−50	216	*67*	8	258	*49*	27	328	*27*	−8	272	*232*	19	266
T2-T0	*91*	−11	316	*33*	−19	262	*48*	11	221	*41*	34	276	*16*	62	348	*193*	−3	265
T3-T0	*67*	53	466	*31*	**150**	**369**	*33*	17	493	*40*	**375**	**623**	*6*	200	371	*156*	**284**	**462**
T4-T0	*40*	**201**	**449**	*29*	**221**	**342**	*28*	12	393	*34*	**268**	**448**	*5*	123	385	*135*	**332**	**509**
**CD4/CD8**																		
T1-T0	*120*	**0.05**	**0.22**	*38*	−0.01	0.2	*67*	−0.03	0.37	*49*	**0.11**	**0.36**	*27*	−0.05	0.37	*232*	0.02	0.23
T2-T0	*91*	**0.08**	**0.26**	*33*	−0.01	0.21	*48*	−0.04	0.5	*41*	**0.10**	**0.19**	*16*	−0.04	0.27	*193*	**0.08**	**0.25**
T3-T0	*67*	**0.11**	**0.32**	*31*	0.02	0.4	*33*	0.13	0.25	*40*	**0.16**	**0.39**	*6*	0.23	0.51	*155*	**0.08**	**0.30**
T4-T0	*40*	0.08	0.45	*29*	0.07	0.41	*28*	0.08	0.25	*33*	**0.16**	**0.28**	*5*	0.34	0.59	*135*	**0.10**	**0.30**

3TC—lamivudine; ABC—abacavir; DTG—Dolutegravir; FTC—emtricitabine; PI—protease inhibitors; RPV—rilpivirine; TAF—tenofovir alafenamide; TDF—tenofovir disoproxil fumarate; T0—baseline; T1—six-month follow-up; T2—one-year follow-up; T3—eighteen-month follow-up; T4—two-year follow-up. Values that significantly changed compared to baseline (*p* < 0.05) are indicated in bold.

**Table 3 viruses-14-02315-t003:** Characteristics of 342 study participants with baseline CD4/CD8 < 1.0, according to the achievement of CD4/CD8 ratio ≥1.0 during the study period.

Variable	CD4/CD8 < 1.0 (n = 241, 70.5%)	CD4/CD8 ≥ 1.0 (n = 101, 29.5%)	*p*
**Baseline CD4/CD8 ratio, mean ± SD**	0.55 ± 0.22	0.75 ± 0.23	<0.0001
**Age, years, mean ± SD**	51.7 ± 10.8	48.8 ± 12.4	0.03
**Sex, n (%)**			
F	55 (62.5)	33 (37.5)	
M	186 (73.2)	68 (67.3)	0.058
**Caucasian ethnicity, n (%)**	220 (71.0)	90 (29.0)	0.53
**Risk factor for HIV acquisition, n (%)**			
Heterosexual	94 (70.2)	40 (29.8)	
MSM	70 (68.6)	32 (31.4)	
IDU	52 (80.0)	13 (20.0)	
Other	25 (61.0)	16 (39.0)	0.19
**HCV coinfection, n (%)**			
HCV RNA −	48 (76.2)	15 (23.8)	
HCV RNA +	20 (87.0)	3 (13.0)	0.30
**HBsAg positive, n (%)**	7 (63.6)	4 (36.4)	0.54
**Time on ART (years), median (IQR)**	10.7 (4.5–18.8)	10.2 (4.4–16.5)	0.55
**CDC stage, n (%)**			
A	93 (63.3)	54 (36.7)	
B	78 (69.6)	34 (30.4)	
C	70 (84.3)	13 (15.7)	0.001
**CD4 (cells/mm^3^), n (%)**			
<250	19 (73.1)	7 (26.9)	
250–499	89 (78.1)	25 (21.9)	
500–749	73 (65.8)	38 (34.2)	
≥750	60 (65.9)	31 (34.1)	0.07
**eGFR (mL/min), n (%)**			
≥90	98 (64.5)	54 (35.5)	
80–89	37 (68.5)	17 (31.5)	
70–79	40 (72.7)	15 (27.3)	
60–69	37 (77.1)	11 (22.9)	
≤60	29 (87.9)	4 (12.1)	0.003
**Regimen, n (%)**			
DTG + 3TC	35 (60.3)	23 (39.7)	
DTG + RPV	19 (70.4)	8 (29.6)	
DTG + PI	35 (83.3)	7 (16.7)	
DTG + TDF/FTC	24 (66.7)	12 (33.3)	
DTG + TAF/FTC	16 (80.0)	4 (20.0)	
DTG + ABC/3TC	112 (70.4)	47 (29.6)	0.20

3TC—lamivudine; ABC—abacavir; BMI—body mass index; CD4—CD4^+^ T cells; CDC—Center for Disease Control and Prevention; CI—confidence interval; DTG—Dolutegravir; eGFR—estimated glomerular filtration rate; FTC—emtricitabine; HCV—hepatitis C virus; IDU—injecting drug use; MSM—men who have sex with men; PI—protease inhibitor; RPV—rilpivirine; TAF—tenofovir alafenamide; TDF—tenofovir disoproxil fumarate.

**Table 4 viruses-14-02315-t004:** Hazard ratios for CD4/CD8 ≥ 1.0 in people living with HIV on 2DR and 3DR DTG-based regimens, according to baseline characteristics.

Variable	Crude HR	95% CI	*p*	Adjusted HR *	95% CI	*p*
**Sex F (ref. M)**	1.55	1.02–2.35	0.04	1.05	0.67–1.65	0.82
**Age (by 1 year)**	0.98	0.96–1.00	0.04			
**Age (ref. < 50 years)**	0.66	0.45–0.97	0.04	0.84	0.55–1.26	0.39
**Risk factor for HIV (ref. hetero)**						
MSM	1.12	0.71–1.79	0.62			
IDU	0.72	0.38–1.35	0.31			
Other	1.12	0.63–2.01	0.69			
**BMI (Kg/m^2^) (by 1)**	1.00	0.95–1.06	0.92			
**BMI (Kg/m^2^) (ref. ≤ 25.0)**						
>25.0–30.0	0.90	0.56–1.44	0.65			
>30.0	1.12	0.58–2.15	0.74			
**Ethnicity (ref. Caucasian)**	1.35	0.72–2.52	0.35			
**HCV coinfection (ref. N)**						
HCV-Ab+/HCV RNA -	0.91	0.53–1.58	0.75			
HCV-Ab+/HCV RNA +	0.36	0.11–1.13	0.08			
**HBsAg positive (ref. HBsAg negative)**	1.26	0.46–3.43	0.65			
**Years of ART (by 1 year)**	1.01	0.99–1.04	0.32			
**CDC stage (ref. A)**						
B	0.90	0.58–1.38	0.63	0.95	0.61–1.48	0.82
C	0.34	0.18–0.62	0.0005	0.44	0.24–0.82	0.009
**CD4 (cells/mm^3^) (ref. < 250)**						
250–499	1.03	0.45–2.39	0.94			
500–749	1.98	0.88–4.44	0.10			
≥750	2.07	0.91–4.71	0.08			
**INSTI in previous regimen**	0.57	0.32–1.00	0.051	0.54	0.30–0.97	0.04
**Baseline CD4/CD8 ratio (by 0.01)**	1.04	1.03–1.05	<0.0001	1.04	1.03–1.06	<0.0001
**eGFR (mL/min) (ref. ≥ 90)**						
80–89	1.01	0.58–1.74	0.98	0.69	0.40–1.21	0.20
70–79	0.79	0.44–1.40	0.42	0.70	0.39–1.26	0.23
60–69	0.59	0.31–1.13	0.11	0.48	0.24–0.93	0.03
≤60	0.35	0.13–0.97	0.04	0.34	0.12–0.96	0.04
**Regimen** (ref. DTG + ABC/3TC)						
DTG + PI	0.82	0.39–1.74	0.60	0.85	0.39–1.84	0.68
DTG + 3TC	1.91	1.15–3.17	0.01	1.34	0.80–2.23	0.27
DTG + RPV	0.74	0.34–1.64	0.46	0.55	0.24–1.54	0.15
DTG + TDF/FTC	0.94	0.50–1.77	0.84	0.92	0.48–1.78	0.79
DTG + TAF/FTC	1.57	0.56–4.40	0.40	0.97	0.34–2.73	0.95

* Variables with *p* < 0.05 were retained in the multivariate model. Sex, age > 50 years, CDC stage, baseline CD4/CD8 ratio (instead of baseline CD4 level), previous INSTI use, and regimen were included. Adjusted HRs for baseline eGFR were calculated including CDC stage and baseline CD4/CD8 ratio (instead of baseline CD4 level). 3TC—lamivudine; ABC—abacavir; BMI—body mass index; CD4—CD4^+^ T cells; CDC—Center for Disease Control and Prevention; CI—confidence interval; DTG—Dolutegravir; eGFR—estimated glomerular filtration rate; FTC—emtricitabine; HCV-Ab—hepatitis C virus antibody; HR—hazard ratio; IDU—injecting drug use; INSTI—integrase strand transfer inhibitors; MSM—men who have sex with men; PI—protease inhibitor; ref—reference; RPV—rilpivirine; TAF—tenofovir alafenamide; TDF—tenofovir disoproxil fumarate.

**Table 5 viruses-14-02315-t005:** Hazard Ratios for CD4/CD8 ≥ 1.0 among PWH on 2DR DTG-based regimens, according to baseline characteristics.

Variable	Crude HR	95% CI	*p*	Adjusted HR *	95% CI	*p*
**Sex F (ref. M)**	1.12	0.56–2.26	0.75			
**Age (by 1 year)**	0.98	0.95–1.01	0.20			
**Age (ref. < 50 years)**	0.68	0.35–1.29	0.23			
**Risk factor for HIV (ref. hetero)**						
MSM	1.62	0.79–3.32	0.19			
IDU	0.51	0.17–1.54	0.23			
Other	0.95	0.32–2.87	0.93			
**BMI (Kg/m^2^) (by 1)**	1.01	0.92–1.11	0.86			
**BMI (Kg/m^2^) (ref. ≤ 25.0)**						
>25.0–30.0	0.95	0.44–2.04	0.89			
>30.0	1.18	0.40–3.56	0.76			
**Ethnicity (ref. Caucasian)**	1.39	0.54–3.57	0.49			
**HCV-Ab- (ref.)**						
HCV-Ab+/HCV RNA −	0.77	0.34–1.74	0.53			
HCV-Ab+/HCV RNA +	n.e.	n.e.	n.e.			
**HBsAg (ref. HBsAg negative)**	1.10	0.15–8.09	0.92			
**Years of ART (by 1 year)**	0.98	0.94–1.02	0.27			
**INSTI in previous regimen**	0.85	0.40–1.80	0.67			
**Baseline CD4/CD8 ratio (by 0.01)**	1.05	1.03–1.08	<0.0001	1.05	1.03–1.08	<0.0001
**CDC stage (ref. A)**						
B	0.73	0.36–1.46	0.37	1.03	0.50–2.11	0.94
C	0.28	0.10–0.82	0.02	0.52	0.18–1.57	0.25
**CD4 (cells/mm^3^) (ref. < 250)**						
250–499	0.28	0.06–1.37	0.12			
500–749	1.03	0.24–4.41	0.97			
≥750	0.88	0.19–4.05	0.87			
**eGFR (mL/min) (ref. ≥ 90)**						
80–89	1.59	0.72–3.52	0.25			
70–79	0.92	0.36–2.35	0.86			
60–69	0.62	0.20–1.86	0.39			
≤60	0.36	0.08–1.55	0.17			
**Regimen (ref. DTG + 3TC)**						
DTG + PI	0.43	0.19–0.99	0.04	0.59	0.25–1.40	0.23
DTG + RPV	0.38	0.16–0.91	0.02	0.38	0.16–0.95	0.04

* variables with *p* < 0.05 were retained in the multivariate model. CDC stage, baseline CD4/CD8 ratio and regimen were included. 3TC—lamivudine; ABC—abacavir; BMI—body mass index; CD4—CD4^+^ T cells; CDC—Center for Disease Control and Prevention; CI—confidence interval; DTG—Dolutegravir; eGFR—estimated glomerular filtration rate; FTC—emtricitabine; HCV-Ab—hepatitis C virus antibody; HR—hazard ratio; ref—reference; TAF—tenofovir alafenamide; TDF—tenofovir disoproxil fumarate, PI—protease inhibitor, RPV—rilpivirine, MSM—men who have sex with men, IDU—injecting drug use.

## Data Availability

The data used and analyzed during the current study are available from the corresponding author on reasonable request.

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
