# Peer review of "Trajectories of CD4+/CD8+ T-Cells Ratio 96 Weeks after Switching to Dolutegravir-Based Two-Drug Regimens: Results from a Multicenter Prospective Cohort Study"

_viruses, 2022, doi:10.3390/v14112315_

Round 1

Reviewer 1 Report

In this paper Taramasso et al, reports a significant increase in the CD4/CD8 ratio in the group of patients who received treatment with DTG/3TC (2DR group) and patients who received DTG +TDF/FTC and DTG + ABC/3TC (3DR) at one year, with a confirmed increase only in patients who received treatment with DTG+TDF/FTC and DTG+ABc/3TC, they conclude that only patients who received treatment with DTG/3TC (2DR) had a significant increase in the CD4/CD8 ratio. They recommend carrying out controlled studies to see the long-term immunological impact of treatment with 2DR regimens.

 Some of these data were reported in the Phase III study TANGO (Clin Infect Dis. 2022 Mar 2;ciac130. doi: 10.1093/cid/ciac130 and Clin Infect Dis.2022 Sep 29;75(6):975-986, doi: 10.1093/cid/ciac036). There were not significant changes in the CD4/CD8 ratio evolution at W48 and 144: Median (IQR) change from baseline to Week 48 and 144 in CD4+/CD8+ cell count ratio was 0.04 (−0.06, 0.13) and 0.06 (−0.06 to 0.20) in the DTG/3TC group and 0.05 (−0.06, 0.13) 0.10 (−0.02 to 0.24) in the CAR group, respectively,

Similar data on the evolution of the CD4/CD8 ratio were also reported in a Spanish prospective cohort of PWH treatment naive who start an ARV with 2DR (DTG/3TC) or 3DR (B/F/T) (Similar CD4/CD8 Ratio Recovery After Initiation of Dolutegravir Plus Lamivudine Versus Dolutegravir or Bictegravir-Based Three-Drug Regimens in Naive Adults With HIV - Front Immunol. 2022; 13: 873408. doi: 10.3389/fimmu.2022.873408).

If the authors wants to evaluate the impact in the CD4/CD8 ratio evolution in all PWH who start all 2DR , it is really of interest at this time  including other regimens different from DTG/3TC or  DTG/RPV?

Regarding the results, in the baseline characteristics of the patients in each group, there is a lack of information on clinical data that could have a clear impact on the evolution of the CD4/CD8 ratio, including a history of virological failure (especially in patients on the DTG/PI group), a history of AIDS-defining diseases and other comorbidities that can have a clear effect on the immune recovery of patients, such as liver cirrhosis.

The tables are very extensive and being able to interpret them can be very tiring, I would recommend simplifying them with the most relevant information of each group, for example:  AIDS diagnosis, Virologic failure during follow-up, etc.

The data that Taramasso et al provide us, could reinforce existing data on the evolution of the CD4/CD8 ratio in patients who start a 2DR, but I would ask them to be able to adjust their results for the for other pathologies that may be present in the patients included in this cohort, such as a history of liver cirrhosis, cancer, as well as other antecedents related to the HIV infection: AIDS-defining diseases, history of virologial failure, etc.

Author Response

REVIEWER1:

In this paper Taramasso et al, reports a significant increase in the CD4/CD8 ratio in the group of patients who received treatment with DTG/3TC (2DR group) and patients who received DTG +TDF/FTC and DTG + ABC/3TC (3DR) at one year, with a confirmed increase only in patients who received treatment with DTG+TDF/FTC and DTG+ABc/3TC, they conclude that only patients who received treatment with DTG/3TC (2DR) had a significant increase in the CD4/CD8 ratio. They recommend carrying out controlled studies to see the long-term immunological impact of treatment with 2DR regimens.

Some of these data were reported in the Phase III study TANGO (Clin Infect Dis. 2022 Mar 2;ciac130. doi: 10.1093/cid/ciac130 and Clin Infect Dis.2022 Sep 29;75(6):975-986, doi: 10.1093/cid/ciac036). There were not significant changes in the CD4/CD8 ratio evolution at W48 and 144: Median (IQR) change from baseline to Week 48 and 144 in CD4+/CD8+ cell count ratio was 0.04 (−0.06, 0.13) and 0.06 (−0.06 to 0.20) in the DTG/3TC group and 0.05 (−0.06, 0.13) 0.10 (−0.02 to 0.24) in the CAR group, respectively,

Similar data on the evolution of the CD4/CD8 ratio were also reported in a Spanish prospective cohort of PWH treatment naive who start an ARV with 2DR (DTG/3TC) or 3DR (B/F/T) (Similar CD4/CD8 Ratio Recovery After Initiation of Dolutegravir Plus Lamivudine Versus Dolutegravir or Bictegravir-Based Three-Drug Regimens in Naive Adults With HIV - Front Immunol. 2022; 13: 873408. doi: 10.3389/fimmu.2022.873408).

If the authors wants to evaluate the impact in the CD4/CD8 ratio evolution in all PWH who start all 2DR , it is really of interest at this time  including other regimens different from DTG/3TC or  DTG/RPV?

RESPONSE:

We thank reviewer 1 for the kind revision and valuable comments.

We know that the other regimens, other than DTG/3TC or DTG/RPV, are less used in clinical practice and, probably, less interesting to readers. However, some people with HIV are forced to do them, and precisely because there are no randomized trials, observational studies, even in small populations, can provide important information. It is clear that patients who are treated with dolutegravir+ PI or other combinations often have a different clinical background than those on DTG/3TC or DTG/RPV, and this is a limitation of the study, because this background itself could be the cause of the poor immune recovery. We have added a disclosure of this limitation in the discussion “Owing to the observational nature of the study, PWH were not randomly assigned to drug therapies, and confounding bias could have occurred, based on clinicians’ decisions to prescribe different drug associations in patients with different baseline characteristics.” Apart from this important limitation, we think these results are important, especially because this observational study also offers the advantage of being prospective and not retrospective.

Regarding the results, in the baseline characteristics of the patients in each group, there is a lack of information on clinical data that could have a clear impact on the evolution of the CD4/CD8 ratio, including a history of virological failure (especially in patients on the DTG/PI group), a history of AIDS-defining diseases and other comorbidities that can have a clear effect on the immune recovery of patients, such as liver cirrhosis.

RESPONSE; We agree with reviewer 1, and we have added in Table 1 data on the prevalence of comorbidities in the study population (liver cirrhosis, malignancies, autoimmune diseases and previous cardiovascular diseases, besides hypertension and diabetes, as well as HCV and HBV co-infections and renal impairment with eGFR <60 ml/min, which were already shown in the previous version;  see Table 1).

The table already reports the number of AIDS-defining conditions in each group (see CDC staging in Table 1).

People with cirrhosis, previous CVD events, autoimmune diseases and/or malignancies were 59. Excluding them from the analysis, we found that the estimated variations from baseline did not vary: they only showed a slight increase, as expected with diminishing sample size, in the standard deviation (in the second decimal point).

In the Results, we added: “After the exclusion of people with comorbidities or conditions potentially affecting the immune recovery (i.e. cirrhosis, autoimmune diseases and/or malignancies, previous CVD events, N= 59), we found that the CD4/CD8 ratios showed similar increases compared to the whole study population”.

Only for the reviewer, we report below data obtained after exclusion of these 59 patients from the analyses:  

1st year

DTG+3TC                                           +0.08 ± 0.27, p=0.005

DTG+ TDF/FTC                                 +0.10 ± 0.20, p=0.004

DTG/3TC/ABC                                   +0.08 ± 0.26, p<0.0001

2nd year

DTG+ TDF/FTC                                 +0.16 ± 0.29, p=0.005

DTG/3TC/ABC                                   +0.10 ± 0.30, p=0.0002

Regarding instead the history of previous virological failures, unfortunately, the protocol of SCOLTA observational cohort does not include information on it. However, we added additional data on past exposure to ART and to the different antiretroviral classes, information that indirectly can give an idea of the patient's previous medical history (See supplementary Table S1).

The tables are very extensive and being able to interpret them can be very tiring, I would recommend simplifying them with the most relevant information of each group, for example:  AIDS diagnosis, Virologic failure during follow-up, etc.

RESPONSE:  we have now simplified Table 1 and 3, removing some information and moving all data regarding previous and current ART exposure to the supplementary materials. There is still a lot of information remaining, but we believe it is necessary for the correct interpretation of the results, as these are characteristics that were different at baseline in the different ART groups (see Table 1).

Only baseline characteristics (and not events occurring during follow-up) are shown in the tables, which is why virologic failures during follow-up are not shown.

The data that Taramasso et al provide us, could reinforce existing data on the evolution of the CD4/CD8 ratio in patients who start a 2DR, but I would ask them to be able to adjust their results for the for other pathologies that may be present in the patients included in this cohort, such as a history of liver cirrhosis, cancer, as well as other antecedents related to the HIV infection: AIDS-defining diseases, history of virological failure, etc.

RESPONSE:

Regrettably, we do not record the history of previous virological failure and all people selected in this analysis were switching from a previous regimen with HIVRNA<50 copies/mL. As suggested, we added in the descriptive tables the history of cirrhosis, cancer, cardiovascular diseases and autoimmune diseases.

The analysis has been repeated after exclusion of study participants with these comorbidities, but confirmed the results of the main analysis (see response to the previous comment).

Reviewer 2 Report

The present work from L. Taramasso and A. Falletta had the aim to evaluate CD4-CD8 dynamics in patients treated with dolutegravir based 2DR and compare them with triple therapies (3DR). It represent a useful study to underlying the importance of reducing number of drugs in the ART regimen.

The authors profoundly explained the results obtained from the study in the tables reported and these values should be helpful for ART management in HIV patients. Nevertheless, in Figure 1, X and Y axis names should be shown. For example “CD4/CD8 ratio” for Y axis and “time” for X axis. In addition, time in X axis should be represented linearly from baseline to 2 years therapy including all time points showed: T0 (baseline), T1 (six months), T2 (one year), T3 (18 months) and T4 (2 years) as now. In this way the follow up would be more clear by considering different points.  

Minor comments:

In Table 1 there is an extra “i” in the column DTG+PI line Others.

In my opinion, the paragraph “5.Conclusions” is not necessary due to a large discussion of the results in the previous paragraph.

Author Response

REVIEWER 2

The present work from L. Taramasso and A. Falletta had the aim to evaluate CD4-CD8 dynamics in patients treated with dolutegravir based 2DR and compare them with triple therapies (3DR). It represents a useful study to underlying the importance of reducing number of drugs in the ART regimen.

The authors profoundly explained the results obtained from the study in the tables reported and these values should be helpful for ART management in HIV patients.

RESPONSE:

We thank reviewer 2 for the kind revision and valuable comments.

 Nevertheless, in Figure 1, X and Y axis names should be shown. For example, “CD4/CD8 ratio” for Y axis and “time” for X axis. In addition, time in X axis should be represented linearly from baseline to 2 years therapy including all time points showed: T0 (baseline), T1 (six months), T2 (one year), T3 (18 months) and T4 (2 years) as now. In this way the follow up would be clearer by considering different points. 

RESPONSE:

Thank you for this suggestion. We have now modified figure 1 in accordance with the reviewer's suggestions, hopefully the figure is now clearer.

Minor comments:

In Table 1 there is an extra “i” in the column DTG+PI line Others.

RESPONSE:

We thank you for noticing this error, which we have now corrected.

In my opinion, the paragraph “5. Conclusions” is not necessary, due to a large discussion of the results in the previous paragraph

RESPONSE:

In accordance with the reviewer’s suggestion, we have removed the “conclusion” paragraph.

Round 2

Reviewer 1 Report

No more comments.